# Backbone: An R package for extracting the backbone of bipartite projections

**Rachel Domagalski**[1], **Zachary P. Neal**[2]*, **Bruce Sagan**[1]

**1** Department of Mathematics, Michigan State University, East Lansing, Michigan, United States of America,
**2** Department of Psychology, Michigan State University, East Lansing, Michigan, United States of America

\* zpneal@msu.edu

## Abstract

Bipartite projections are used in a wide range of network contexts including politics (bill co-sponsorship), genetics (gene co-expression), economics (executive board co-membership), and innovation (patent co-authorship). However, because bipartite projections are always weighted graphs, which are inherently challenging to analyze and visualize, it is often useful to examine the 'backbone,' an unweighted subgraph containing only the most significant edges. In this paper, we introduce the R package `backbone` for extracting the backbone of weighted bipartite projections, and use bill sponsorship data from the 114th session of the United States Senate to demonstrate its functionality.

## Introduction

Networks are useful for studying many different phenomena in the natural and social worlds, but network data can be difficult to collect directly. As a result, it is common for research to measure an unobserved unipartite network of interest using a bipartite projection in which the edges capture whether (or the extent to which) two vertices co-participate in a relevant event. For example, friendship networks are measured using event co-attendance [1], political networks are measured using bill co-sponsorship [2], executive networks are measured using board co-membership [3], scholarly collaboration networks are measured using paper co-authorship [4], knowledge networks are measured using paper co-citation [5], and genetic networks are measured using gene co-expression [6]. Indeed, some have gone so far as to argue that "every one-mode network can be regarded as a projection of a bipartite network" [7, see also [8, 9]].

However, there are several challenges to studying bipartite projections. Two of these challenges arise from the projection function itself. First, the projection function "transforms the problem of analysing a bipartite structure into the problem of analysing a weighted one, which is not easier" [10, p. 34-35]. Second, bipartite projections introduce topological characteristics such as inflated clustering, such that the projection of "even a random [bipartite] network—one that has no particular structure built into it at all—will be highly clustered" [11, p. 128].

Two additional challenges arise from characteristics of the bipartite data, which are usually lost in the projection transformation. First, when transforming a bipartite graph into a unipartite graph via projection, information about the artifacts responsible for edges between vertices

**Data Availability Statement:** The data and code necessary to replicate the examples in this paper are available at https://osf.io/myje5/.

**Funding:** ZN and BS received funding from the National Science Foundation (#1851625 &

#2016320). The funders had no role in study design, data collection and analysis, decision to publish, or preparation of the manuscript.

**Competing interests:** The authors have declared that no competing interests exist.

is lost [10], in particular, one no longer knows *which* artifact(s) gave rise to a given edge and therefore no longer knows whether the artifact(s) are large or small (i.e. the column sums of the bipartite matrix). This is important because co-participation in large artifacts provides more information about the relationship between two vertices than co-participation in small artifacts [12]. For example, observing two people attending the same small party provides more information about a potential social relationship between them than observing these individuals attending the same large gathering. Similarly, observing two legislators co-sponsoring the same unpopular bill (i.e. one that is co-sponsored by no one else) provides more information about a potential political relationship between them than observing these legislators co-sponsoring the same popular bill (i.e. one that is co-sponsored by many others also).

Second, bipartite projection also involves the loss of information about the individual vertices, in particular, one no longer knows *how many* artifacts a given vertex participated in (i.e. the row sums of the bipartite matrix). This information is important to consider because the scale of each edge weight in a bipartite projection is driven by the number of artifacts participated in by the two vertices it connects [12]. For example, on average the number of events co-attended by two people who each attend many events will be larger (on average) than the number of events co-attended by two people who each attend few events. Similarly, on average the number of bills co-sponsored by two legislators who each sponsor many bills will be larger (on average) than the number of bills co-sponsored by two legislators who each sponsor few bills. Therefore, what counts as a 'large' or 'small' number of co-attendances or co-sponsorships depends in part on the total number of attendances or sponsorships of both members of a dyad. As we will see, the backbone extraction methods we consider cope with these challenges by controlling for the row and column sums of the bipartite matrix associated with the bipartite graph in question.

For these reasons, it is often useful to extract and study the *backbone* of a bipartite projection. The backbone is an unweighted (i.e. an edge is either present or absent) or signed (i.e. an edge can be positive, negative, or absent) graph that preserves only the most "significant" edges from the weighted bipartite projection. Multiple methods of backbone extraction exist for bipartite projections [12], but until now there has not been a single software package that implements these methods. In this paper, we introduce and demonstrate version `1.2.2` of the `backbone` R package, which implements four backbone extraction methods—a universal threshold, a hypergeometric model, a stochastic degree sequence model, and a fixed degree sequence model—in a common framework that facilitates their use. It is possible to install the package in R [13] from the Comprehensive R Archive Network (CRAN), load it for use, and verify its version number using:

```
> install.packages("backbone")
> library(backbone)
> sessionInfo()[["otherPkgs"]][["backbone"]][["Version"]]
[1] "1.2.2"
```

Further information regarding the CRAN distribution is available at https://CRAN.R-project.org/package=backbone. Additional materials relating to `backbone`, including papers, presentations, workshop materials, and data sets are available at http://www.zacharyneal.com/backbone. Replication data and code for the examples presented below are available at https://osf.io/myje5/.

## Graph theory preliminaries

A *graph G* is a set of objects called *vertices*, together with a set of 2-element subsets of the vertices which are called *edges*. An edge between vertices $i$ and $j$ can be denoted as $e = ij$. If there

exists an edge $e = ij$ between vertices $i$ and $j$, we say that $i$ and $j$ are *adjacent*. The *degree* of vertex $i$ is the number of edges of the form $ij$ for some $j$. The *adjacency matrix* of a graph $G$ with $n$ vertices is an $n \times n$ matrix $G = [G_{ij}]$ where $G_{ij} = 1$ if an edge is present between vertex $i$ and vertex $j$, or $G_{ij} = 0$ if that edge is absent. We call a graph *weighted* if each edge has an associated numeric value, and unweighted otherwise. In the weighted case $G_{ij} = w(ij)$ where $w(ij)$ is the weight of the edge $e = ij$. We make no distinction between a graph and its adjacency matrix.

We call a graph *bipartite* if the set of vertices can be partitioned into two sets $U$ and $W$ such that each edge of the graph is of the form $ij$ where $i \in U$ and $j \in W$. The sets $U$ and $W$ are called *independent*, meaning there are no edges present within the sets. These graphs can be represented by a *biadjacency* matrix, $B$. This matrix uses rows to represent vertices in the set $U$ and columns to represent vertices in the set $W$. We set $B_{ij} = 1$ if there is an edge between vertex $i$ of $U$ and vertex $j$ of $W$, and set $B_{ij} = 0$ otherwise. As with graphs, we conflate a bipartite graph with its biadjacency matrix. The row sum of the $i$th row of the biadjacency matrix, denoted $R_i$, is equal to the degree of the $i$th vertex in the set $U$. Similarly, the column sum of the $j$ column of $B$, denoted $C_j$, is the degree of the $j$th vertex in $W$.

We can use bipartite graphs to study social networks where the vertices of $U$ are agents (e.g., people), the vertices of $W$ are artifacts (e.g., events), and edges represent the affiliation of a person with an artifact. These bipartite networks are often also called affiliation networks or two-mode networks. To transform this data into a weighted graph, we *project* the bipartite adjacency matrix $B$ by multiplying it by its transpose $B^\top$. This produces a weighted graph $G = BB^\top$ called the *bipartite projection*, where $G_{ij}$ is equal to the number of artifacts of $W$ with which both $i$ and $j$ are affiliated when $i \neq j$. The value $G_{ii}$ is equal to the total number of artifacts with which $i$ is affiliated. Let

$$M_{ij} = \min(G_{ii}, G_{jj}) - (|W| - \max(G_{ii}, G_{jj})).$$

The value of each off-diagonal entry $G_{ij}$ is bounded by

$$\max(0, M_{ij}) \leq G_{ij} \leq \min(G_{ii}, G_{jj}).$$

Bipartite projections are of interest in social network analysis because they allow us to construct a network from artifact affiliations, which are often easier to obtain than taking a survey of the network members. If the number of members of the network is large, getting complete and reliable information regarding relationships between members can involve long and repetitive survey techniques which can lead to survey fatigue and costly field work. Additionally, in some settings, individuals may be reluctant to provide information about their relationships. Moreover, in historical contexts, a survey of long-deceased network members is simply not possible. In these cases, it is often beneficial and easier to collect bipartite network information, such as event attendance, then project to infer a weighted network between the social figures.

Because bipartite projections are weighted, and because what counts as a 'large' or 'small' weight can differ for each edge, it can be useful to reduce this information by focusing on an unweighted subgraph that contains only the most important edges. We call this subgraph a *backbone* of $G$, which we denote as $G'$.

## Backbone extraction methods

The simplest approach to extracting a backbone of bipartite projections applies a single, universal threshold $T$ to all edges such that:

$$G'_{ij} = \begin{cases} 1 & \text{if } G_{ij} > T \\ 0 & \text{if } G_{ij} \leq T \end{cases}.$$

For example, a threshold value of $T = 0$ indicates that as long as a pair of vertices are affiliated with at least one of the same artifacts (i.e. they have a nonzero edge weight), an edge between them will be present in the backbone. We call this type of backbone *binary*. It is also possible to extract a *signed* backbone by selecting distinct upper $T^+$ and lower $T^-$ thresholds, $T^- < T^+$, such that

$$G'_{ij} = \begin{cases} 1 & \text{if } G_{ij} > T^+ \\ -1 & \text{if } G_{ij} < T^- \\ 0 & \text{if } T^- \leq G_{ij} \leq T^+ \end{cases}.$$

However, backbones extracted using universal thresholds are very dense and clustered, and therefore tend to be uninformative [10, 12].

An alternative approach to backbone extraction, and the approach that `backbone` is designed to facilitate, identifies important edges to be retained in the backbone using a statistical test that compares an edge's observed weight to the distribution of its weights under a null model. Given a null model that can generate a distribution of an edge's weights $G^*_{ij}$, it is possible to compute the probability that an edge's observed weight is in the upper ($P(G^*_{ij} \geq G_{ij})$) or lower ($P(G^*_{ij} \leq G_{ij})$) tail of this distribution, and therefore not likely to be the result of such a null model. Using these probabilities and a significance level $\alpha$ for the test, a *signed* backbone can be extracted such that:

$$G'_{ij} = \begin{cases} 1 & \text{if } P(G^*_{ij} \geq G_{ij}) < \alpha/2 \\ -1 & \text{if } P(G^*_{ij} \leq G_{ij}) < \alpha/2 \\ 0 & \text{otherwise} \end{cases}.$$

In many cases, a simpler *binary* backbone may be more useful, which can be achieved by discarding the negative edges such that:

$$G'_{ij} = \begin{cases} 1 & \text{if } P(G^*_{ij} \geq G_{ij}) < \alpha/2 \\ 0 & \text{if } P(G^*_{ij} \geq G_{ij}) \geq \alpha/2 \end{cases}.$$

However, whether a binary or signed backbone is extracted, a two-tailed significance test is used because, for any given edge, the observed value in the projection could be in either tail of the null distribution.

The challenge to using the statistical approach to backbone extraction lies in defining a suitable null model and computing the required probabilities (i.e. edge-wise *p*-values). A family of nine null models can be defined by the constraints they place on the row and column sums in a random bipartite graph $B^*$: The row sums in $B^*$, and separately the column sums in $B^*$, can be unconstrained, constrained to match those in $B$ *on average*, or constrained to match those in $B$ *exactly* [14, 15]. Let $\mathcal{R}$ be a set of restrictions on the row and column vertex degrees

corresponding to one of these nine possibilities, and let $\mathcal{B}(\mathcal{R})$ be the space of all bipartite graphs $B^*$ which satisfy those conditions. This approach to backbone extraction compares the values $G_{ij}$, the bipartite projection of interest, to the distributions that describe $G^*_{ij} = (B^*B^{*\top})_{ij}$ for all bipartite graphs $B^* \in \mathcal{B}(\mathcal{R})$. Here, we focus on three sets of restrictions $\mathcal{B}(\mathcal{R})$, defining three distinct null models: (1) the hypergeometric model, in which row sums are constrained to match those in $B$ exactly, but column sums are unconstrained; (2) the stochastic degree sequence model, in which both row and column sums are constrained to match those in $B$ on average; and (3) the fixed degree sequence mode, in which both row and column sums are constrained to match those in $B$ exactly.

## Hypergeometric Model (HM)

The hypergeometric model constrains the row sums in $B^*$ as equal to their values in $B$ (i.e. fixed), but places no constraints on the column sums [16, 17]. The distribution of $(B^*B^{*\top})_{ij}$ for all bipartite graphs $B^* \in \mathcal{B}(\mathcal{R})$ is given by the hypergeometric distribution.

The hypergeometric distribution is a discrete probability distribution that models the probability of having $k$ "successes" in a random sample of size $n$ (drawn without replacement) from a population of size $N$, where $K$ of the objects are considered successes. The probability mass function is given by

$$P(X = k) = \frac{\binom{K}{k}\binom{N-K}{n-k}}{\binom{N}{n}}$$

where $X$ is the random variable of the distribution.

In the case of a bipartite projection, let $B^*$ be a bipartite graph with independent sets $U$ and $W$. Let $U = \{u_1, u_2, \ldots, u_m\}$. We denote the neighborhood of a vertex $i$ by $N(i)$, meaning the set of vertices to which $i$ is adjacent. Then entry $G^*_{ij} = (B^*B^{*\top})_{ij}$ is the number of vertices in $N(u_i)$ which are also in $N(v_j)$. So the corresponding distribution is hypergeometric with population $N = |W|$, sample size $n = |N(u_i)|$, $K = |N(u_j)|$ possible "successful" objects, and $k = |N(u_i) \cap N(u_j)|$ successes.

This method ensures that both $i$ and $j$ are affiliated with the same number of artifacts we've observed in the original data, but the number of individuals that are affiliated with each artifact may vary. From the probability mass function we can find the probability of $i$ and $j$ participating in at least $G_{ij}$ events (for positive backbone edges) or at most $G_{ij}$ events (for negative backbone edges).

## Stochastic Degree Sequence Model (SDSM)

The stochastic degree sequence model constrains $B^*$ so that its *expected* row sums and *expected* column sums equal those observed in $B$ [12]. That is, for any given $B^*$, each row and column sum may be higher or lower than what is observed in $B$, but if one takes the average over all the possible $B^*$ of the sum of a given row or column then one obtains the corresponding sum in $B$. When $B^*$ is generated by filling the $B^*_{ij}$ with the outcomes of independent Bernoulli trials, the distribution of $(B^*B^{*\top})_{ij}$ for all bipartite graphs $B^* \in \mathcal{B}(\mathcal{R})$ is given by a Poisson binomial distribution [18]. A *Bernoulli trial* is a random variable with exactly two outcomes, often referred to as "success" and "failure", or "1" and "0." The Poisson binomial distribution is the distribution of a sum of independent Bernoulli trials. We let $p_k$ be the probability of success of the $k$th trial and call the $p_k$ the *parameters* of the distribution. When the probability of success on each Bernoulli trial is equal, this reduces to the ordinary binomial distribution.

The Poisson binomial distribution models the probability of getting $k$ successes in $n$ trials, where each trial has a specific probability for success [18, 19]. To apply the Poisson binomial distribution to compute $P(G_{ij}^* \geq G_{ij})$ we must first determine the independent probabilities $P(B_{ij}^* = 1)$. There are several ways to determine $P(B_{ij}^* = 1)$, including simple equations such as $(R_i \times C_j) \backslash \sum B$ where $\sum B$ is the sum of each entry of the biadjacency matrix of $B$ [15, 18], and predicted probabilities from binary outcome models [12]. These older methods are available in the `backbone` package, but here we introduce and describe the *polytope* method [20].

The polytope method revolves around creating a convex set. A subset $C$ of $\mathbb{R}^n$ is *convex* if for any $a$, $b \in C$, the line segment joining $a$ and $b$ is also in $C$. The *convex hull* of a set points $A \subseteq \mathbb{R}^n$, $conv(A)$, is the smallest convex set containing $A$. When $A$ is a finite set, the convex hull of $A$ is called the *polytope* generated by $A$.

Consider all $m \times n$ matrices in $\mathcal{B}(\mathcal{R})$ as vectors in $\mathbb{R}^{mn}$, where $\mathcal{B}(\mathcal{R})$ is all zero-one matrices with the same row and column sums as $B$. Let this set of vectors be $A$ and consider the convex hull of $A$. As matrices, we see that $conv(A)$ is the set of matrices in $\mathbb{R}^{m \times n}$ with the same row and column sums as $\mathcal{B}(\mathcal{R})$, with all entries constrained between 0 and 1.

From this set of matrices, we find the matrix $M_{max}$ that maximizes the entropy function

$$H(M) = \sum_{i,j} \left( M_{ij} \ln \frac{1}{M_{ij}} + (1 - M_{ij}) \ln \frac{1}{1 - M_{ij}} \right)$$

on $conv(A)$. The function $H$ is strictly concave and $conv(A)$ is a convex set which guarantees a unique solution to the maximization. The matrix $M_{max}$ is then used as a matrix of probabilities where $P(B_{ij}^* = 1)$ is equal to $(M_{max})_{ij}$.

Once we have the matrix of probabilities $P_{ij} = P(B_{ij}^* = 1)$ we note that, since $G^* = B^* B^{*\top}$,

$$G_{ij}^* = B_{i1}^* B_{j1}^* + B_{i2}^* B_{j2}^* + \cdots + B_{in}^* B_{jn}^*.$$

Furthermore, since the probabilities are independent,

$$P(B_{ik}^* B_{jk}^* = 1) = P(B_{ik}^* = 1)P(B_{jk}^* = 1) = P_{ik} P_{jk}.$$

It follows that $G_{ij}^*$ is Poisson binomial with parameters $p_1 = P_{i1} P_{j1}, \ldots, p_n = P_{in} P_{jn}$.

Computing the exact cumulative distribution function for a Poisson binomial distribution is difficult. However, the Poisson binomial distribution is well-approximated by a refined normal distribution [19], which we use in the stochastic degree sequence model.

## Fixed Degree Sequence Model (FDSM)

The fixed degree sequence model constrains both the row and column sums in $B^*$ to be equal to their values in $B$ (i.e. both are fixed) [21]. The probability distribution that describes $(B^* B^{*\top})_{ij}$ for all bipartite graphs $B^* \in \mathcal{B}(\mathcal{R})$ is unknown, and thus an approximate distribution is constructed via simulation:

1. Construct a bipartite graph $B^*$ that represents a random draw from $\mathcal{B}(\mathcal{R})$.

2. Project $B^*$ (i.e. compute $B^* B^{*\top}$) to obtain a random weighted bipartite projection $G^*$.

3. Repeat steps 1 and 2 $N$ times to sample the space of possible $G_{ij}^*$.

The matrix multiplication required in step 2 is computationally expensive but straightforward [22]. However, the random sampling of a $B^*$ from $\mathcal{B}(\mathcal{R})$ in step 1 is more challenging. Several methods have been suggested, including Markov Swap methods [23] and Sequential Importance Sampling [24], but in the fixed degree sequence model we use the curveball

algorithm [25], which is one of the fastest algorithms that has been proven to randomly sample [[26, see also [27]].

## Applying `backbone` to the US Senate

We illustrate the use of the R `backbone` package to extract the backbone of a network of bill co-sponsorship relations among Senators in the 114[th] session of the United States Senate. This context provides insight into how the `backbone` package works because both prior research [2, 28–31] and media accounts [32] of the current US political climate provide us with *a priori* expectations about what structure a properly extracted backbone should have. Specifically, given conditions of partisanship and polarization, we should expect to see positive relationships form primarily between those in the same political party, and accordingly a relatively large modularity statistic computed from a partition of the network's nodes by political party. In visualizations of the extracted backbones, we depict Republican senators by red vertices, and both Democratic and Independent senators who are left-leaning and caucused with Democrats by blue vertices. Although we discuss signed backbones in the text, for visual clarity we only provide figures for binary backbones which contain positive edges. Positive relations of collaboration between two Republicans are red, between two Democrats are blue, and for all other pairs are purple. Replication data and code are available at https://osf. io/myje5.

### Data

The data consists of 100 senators and the 3589 bills that they have sponsored or co-sponsored in the 114[th] session of Congress [33]. This data takes the form of a bipartite graph *B*, where the two independent sets of vertices are the senators (agents) and the bills (artifacts). Each edge connects one senator to one bill. Specifically, $B_{ij} = 1$ if senator *i* sponsored or co-sponsored bill *j*, and otherwise is 0. Below we examine the data set. Notice that the row names correspond to each senator (including their party affiliation and the state they represent) and the column names refer to the bill number.

```
> set.seed(19)
> library(backbone)
> senate <- read.csv("S114.csv", row.names = 1, header = TRUE)
> senate <- as.matrix(senate)
> dim(senate)
[1] 100 3589
> senate[1:5, 1:5]
                        sj9 sj8 sj7 sj6 sj5
Alexander, L. (TN-R)      0   1   0   1   0
Boxer, B. (CA-D)          0   0   0   0   1
Cantwell, M. (WA-D)       0   0   0   0   1
Carper, T. (DE-D)         0   0   0   0   1
Cochran, T. (MS-R)        0   1   0   1   0
```

A weighted graph *G* can be constructed from *B* via bipartite projection, where $G = BB^{\top}$ and $G_{ij}$ contains the number of bills that both senator *i* and senator *j* sponsored. Notice the graph is now 100 rows by 100 columns.

```
> G <- senate%*%t(senate)
> dim(G)
[1] 100 100
> G[1:5, 1:2]
```

```
                   Alexander, L. (TN-R) Boxer, B. (CA-D)
Alexander, L. (TN-R)              141               10
Boxer, B. (CA-D)                   10              303
Cantwell, M. (WA-D)                15               82
Carper, T. (DE-D)                  12               55
Cochran, T. (MS-R)                 40               25
```

The projected graph *G* now indicates that Senator Lamar Alexander sponsored a total of 141 bills in the 114[th] session. Among these 141 bills, 10 were co-sponsored with Senator Barbara Boxer, and 15 were co-sponsored with Senator Maria Cantwell.

We can use the values of graph *G* to observe differences between those with similar or dissimilar ideology. Below, we compare the number of bills co-sponsored by two individuals with similar political ideology, Senators Cory Booker and Elizabeth Warren, versus those with dissimilar ideology, Senators Ted Cruz and Bernie Sanders. The results are consistent with the expectation that legislators sharing a similar ideology engage in more co-sponsorships.

```
> G["Booker, C. (NJ-D)", "Warren, E. (MA-D)"]
[1] 98
> G["Cruz, T. (TX-R)", "Sanders, B. (VT-I)"]
[1] 5
```

The differences in the number of bills co-sponsored prompts an important underlying question: how many bills do two senators have to co-sponsor before we would be justified in concluding they are political collaborators? Similarly, how few bills do they have to co-sponsor before we would be justified in concluding they are political opponents? These questions are what the `backbone` package seeks to answer.

## Extracting a universal threshold backbone: Universal()

The simplest approach to backbone extraction applies a single threshold *T* to all edges. A threshold value of 0 indicates that as long as a pair of vertices are affiliated with at least one of the same artifacts (i.e. they have a nonzero edge weight), their relationship should be counted. However, this can lead to extremely dense and uninformative networks, as we will show with our example data. In any case, if using a single threshold value *T* is desired, this can be done in the `backbone` package by using the `universal()` function. This function allows the user to extract an unweighted backbone by selecting a single threshold *T*, or extract a signed backbone by selecting upper and lower thresholds $T^+$ and $T^-$.

Using the `senate` data set, we use the `universal()` function to compute a backbone with a single threshold of 0. Thus if two senators have co-sponsored one or more bills, there will be an edge between them. Notice that our backbone graph is represented by a square adjacency matrix with 0-1 entries.

```
> universalbb1 <- universal(senate, upper = 0,
bipartite = TRUE)
> universalbb1$backbone[1:5, 1:2]
                   Alexander, L. (TN-R) Boxer, B. (CA-D)
Alexander, L. (TN-R)                0                1
Boxer, B. (CA-D)                    1                0
Cantwell, M. (WA-D)                 1                1
Carper, T. (DE-D)                   1                1
Cochran, T. (MS-R)                  1                1
```

This universal threshold backbone requires approximately 0.01 seconds to extract (all running times are reported for a 2.3 Ghz Quad-Core Intel Core i7). Plotting this backbone using

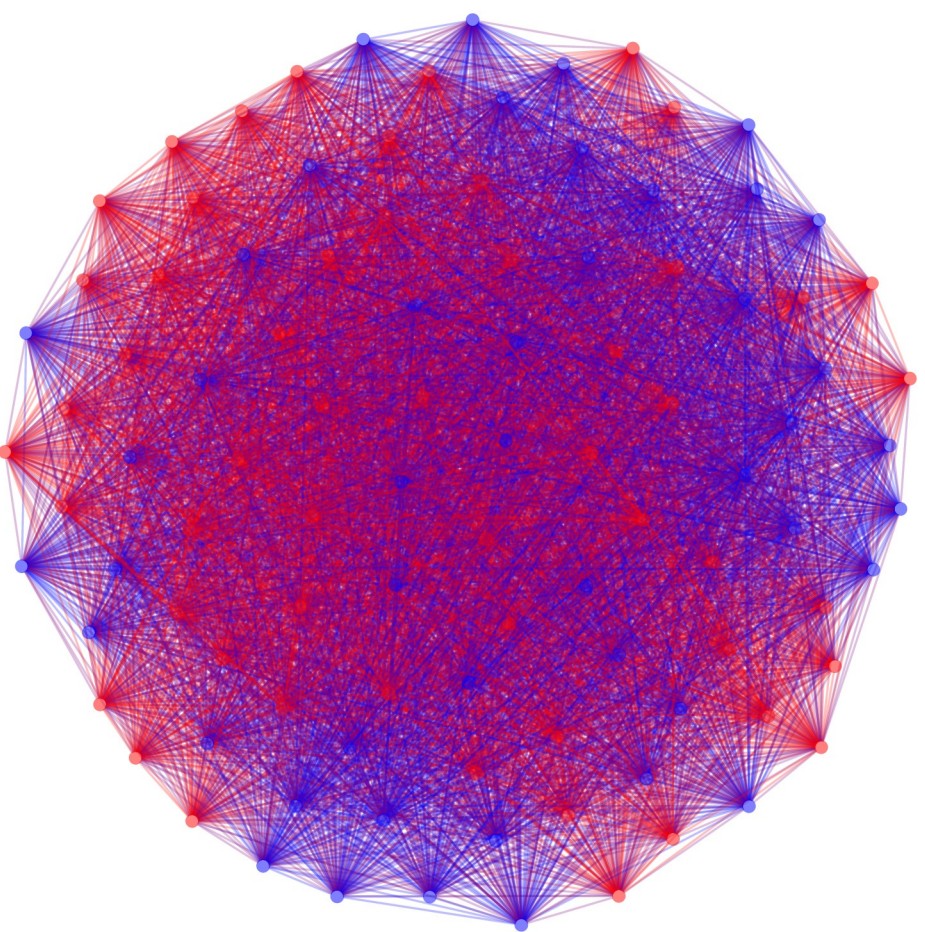

**Fig 1. The positive backbone of the US Senate co-sponsorship network with edges retained between two senators if they sponsored at least 1 bill together.**

the `igraph` package [34] reveals that it is extremely dense as only 1 pair of senators out of the total 4950 unique pairs have not sponsored at least one bill together (see Fig 1). Accordingly, this universal threshold backbone is uninformative about the underlying structure of the network. Moreover, partitioning this backbone into two groups by political party yields a modularity near zero, which indicates that this backbone does not reflect the partisan polarization known to exist in the US Senate.

To create a signed backbone, we can apply both an upper and lower threshold value. The following code will return a backbone where the positive edges indicate two senators co-sponsored more than 1 standard deviation above the mean number of co-sponsored bills and negative edges indicate two senators co-sponsored less than 1 standard deviation below the mean number of co-sponsored bills. The graph of the positive edges of this backbone can be seen in Fig 2.

```
> universalbb2 <- universal(senate, upper = function(x) mean
(x)+sd(x),
                    lower = function(x) mean(x)-sd(x), bipar-
tite = TRUE)
```

This universal threshold backbone requires approximately 0.01 seconds to extract. The resulting graph in Fig 2 is much less dense than when using an upper threshold of 0.

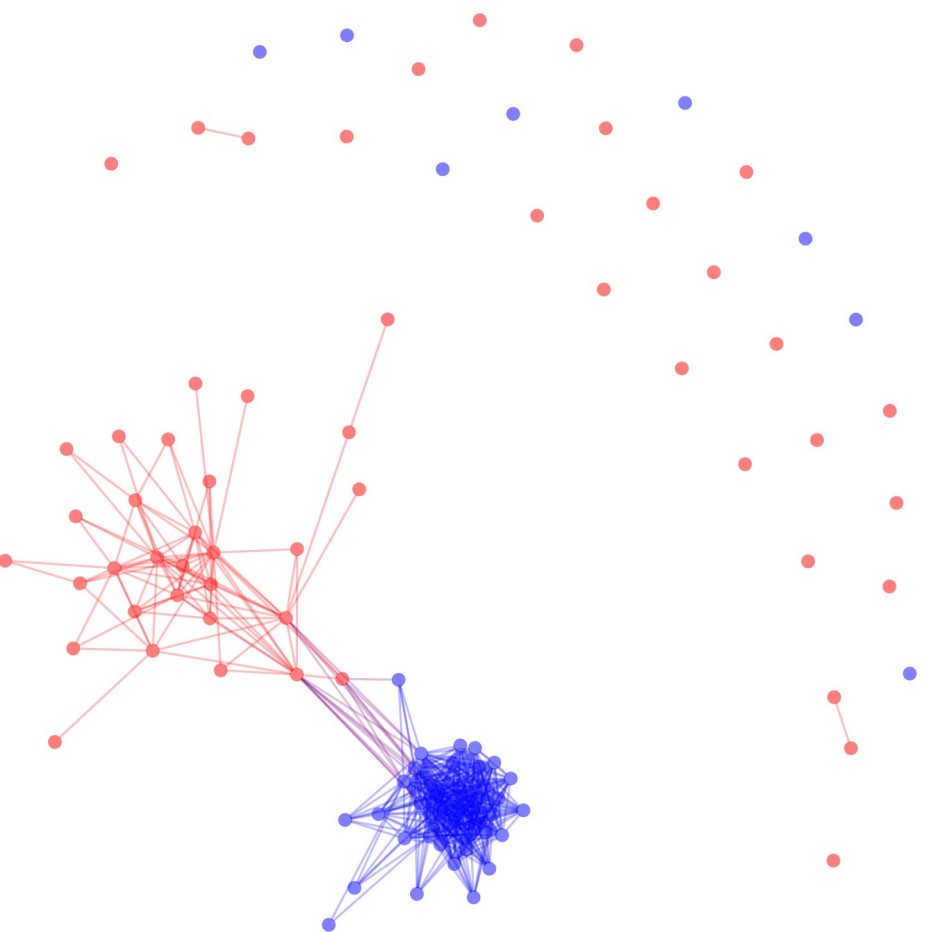

**Fig 2. The positive backbone of the US Senate co-sponsorship network with edges retained between two senators if they sponsored more bills together than one standard deviation above the mean.**

Additionally, the polarized structure of the Senate by political party is visible, and is confirmed by a larger modularity ($M = 0.277$). However, it still does not necessarily reveal the underlying structure of the network among legislators. In this case, "the application of a threshold to the global weight distribution. . .belittles nodes with a small [degree]," resulting in a backbone that preserves edges only among legislators who sponsor many bills, and treating legislators who sponsor few bills as isolates [35, p. 6484]. To obtain meaningfully sparse graphs that do not ignore the multi-scalar character of node degrees we must allow the threshold to vary for different edges. To improve our backbone results, we move to methods of bipartite projection backbones that rely on a distinct threshold value for each pair of vertices.

## Extracting a null model backbone: Backbone.extract()

Instead of using a universal threshold to determine a backbone, the `backbone` package permits using a model that is based on a statistical test, such as the hypergeometric model, stochastic degree sequence model, or fixed degree sequence model. To use these methods in `backbone`, one first calls to a null model function (`hyperg()`, `sdsm()`, or `fdsm()`) which finds the probability of observing an edge with the observed weight in a corresponding null model, returning an object of class 'backbone.' This object is then supplied to

`backbone.extract()`, which performs the hypothesis test for a given significance value and returns a backbone graph. The user can input bipartite graph objects of class 'matrix', 'sparseMatrix', 'Matrix', 'igraph', 'network', and 'edgelist' (a matrix of two columns), and can choose the type of backbone returned by specifying the desired class in `backbone.extract()`.

The `backbone.extract()` function allows the user to input the backbone class object and obtain either a signed or positive backbone. The `backbone.extract()` function has five arguments: `matrix`, `signed`, `alpha`, `class`, `narrative`, and `fwer`. The `matrix` argument takes a backbone object generated by `hyperg()`, `sdsm()`, or `fdsm()` and returns a backbone graph of class = `class` using a two-tailed significance test with significance value $\alpha$ = alpha. If the `signed` parameter is set to `TRUE` then a signed backbone is returned, if it is set to `FALSE` then a positive backbone is returned. If the `narrative` parameter is set to `TRUE` then suggested narrative text for a manuscript, including possible citations, is displayed.

Extracting the backbone of a bipartite projection involves conducting an independent statistical test on $\ell = m(m - 1)/2$ edges in the projection, where $m$ is the number of vertices in the bipartite projection. Because each of these tests is independent, this can inflate the familywise error rate beyond the desired `alpha`. The `fwer` parameter offers two ways to correct for this: the classical Bonferroni correction is applied when `fwer = 'bonferroni'`, and the more powerful Holm-Bonferroni correction is applied when `fwer = 'holm'` [36].

**Hypergeometric backbone: Hyperg().** To apply the hypergeometric distribution to a bipartite graph, one uses the `hyperg()` function. The `hyperg()` function returns a backbone class object that contains: a `positive` matrix with $(i, j)$ entry equal to the probability that $G_{ij}^*$ is equal to or above the corresponding entry in $G$, and a `negative` matrix with $(i, j)$ entry equal to the probability that $G_{ij}^*$ is equal to or below the corresponding entry in $G$, and a `summary` data frame with includes the name of the model, number of rows and columns in the adjacency matrix of the input graph, skew of the row and columns, and running time of the model.

```
> hypergprobs <- hyperg(senate)
Finding the distribution using hypergeometric distribution
> hypergbb <- backbone.extract(hypergprobs, alpha = .01)
```

The hypergeometric backbone requires approximately 0.02 seconds to extract. We can now examine how this method has changed the appearance of our network, focusing only on the positive edges of the signed backbone in Fig 3. We can see that the hypergeometric function has reduced the density of our network and that we begin to see some of the two party structure that is inherent in the United States Senate. The known polarized structure is also apparent, which is reflected in this network's modularity ($M = 0.215$).

Specifically, for our example, the hypergeometric function will fix the number of bills that each senator sponsors, while allowing each bill to be sponsored by a varying number of senators. The function will compute the probability of each senator sponsoring at least (or at most) the observed number of bills when the bills which they sponsor were chosen randomly.

**The stochastic degree sequence model: Sdsm().** The `sdsm()` function returns a backbone object containing the same objects as `hyperg()`: matrices `positive` and `negative` containing probabilities under the stochastic degree sequence model, as well as a `summary` data frame.

In the context of the senate co-sponsorship matrix, the stochastic degree sequence model will compare our observed values to a distribution where each senator sponsors roughly the same number of bills, and each bill is sponsored by roughly the same number of people.

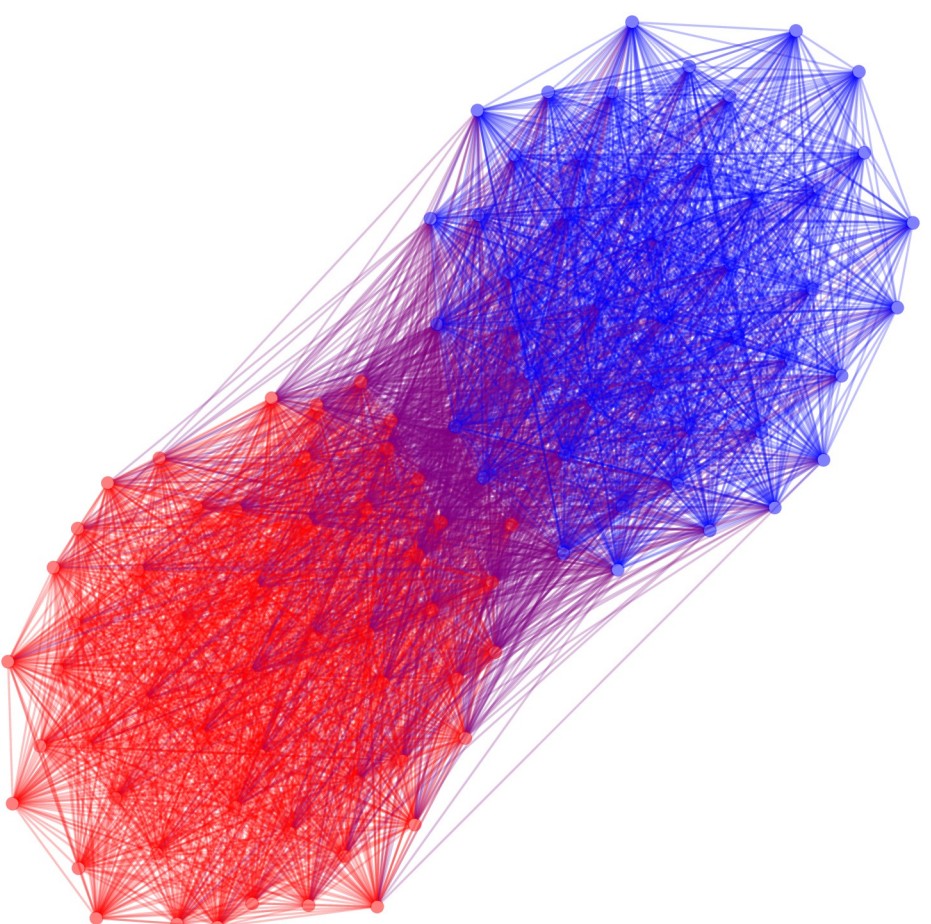

**Fig 3. The positive backbone of the US Senate co-sponsorship network under the hypergeometric model.**

Here we use the `polytope` model, the default in `backbone v1.2.2`, to compute the probabilities for the Poisson binomial distribution.

```
> sdsm <- sdsm(senate)
Finding the distribution using SDSM with polytope model.
> sdsmbb <- backbone.extract(sdsm, narrative = TRUE, alpha =
.01)
Suggested manuscript text and citations:
From a bipartite graph containing 100 agents and 3589 arti-
facts, we obtained the weighted bipartite projection, then
extracted its signed backbone using the backbone package (Doma-
galski, Neal, & Sagan, 2021). Edges were retained in the back-
bone if their weights were statistically significant
(alpha = 0.01) by comparison to a null Stochastic Degree
Sequence Model (Neal, 2014).
Domagalski, R., Neal, Z. P., and Sagan, B. (2021). backbone:
An R Package for Backbone Extraction of Weighted Graphs. PLoS
ONE.
Neal, Z. P. (2014). The backbone of bipartite projections:
Inferring relationships from co-authorship, co-sponsorship,
```

co-attendance and other co-behaviors. Social Networks, 39, 84-
97. https://doi.org/10.1016/j.socnet.2014.06.001

The SDSM backbone requires approximately 45 seconds to extract using the polytope model. We are able to see more of the partisan structure that is suggested to be present in the US Senate in Fig 4, and this visualization provides more information than the extremely dense graph found using a universal threshold. Moreover, the known polarized structure of the US Senate is particularly evident, and confirmed by the much larger modularity ($M = 0.471$).

**The fixed degree sequence model: Fdsm().** The fixed degree sequence model first constructs a random bipartite graph $B^*$ that preserves both degree sequences using the curveball algorithm [14]. This bipartite graph $B^*$ is then projected to obtain a random weighted bipartite projection $G^* = B^* B^{*\top}$. These two steps are repeated a number of times to sample the space of possible $G^*_{ij}$. At each iteration, we compare $G_{ij}$ to the value of $G^*_{ij}$ and keep a record of how often it was above, below, or equal to the generated value. The fdsm() function returns a backbone object containing a matrix object positive of the proportion of times $G^*_{ij}$ is equal to or above the corresponding entry in $G$, and a matrix object negative containing the proportion of times $G^*_{ij}$ is equal to or below the corresponding entry in $G$, and a summary data frame as in hyperg() and sdsm().

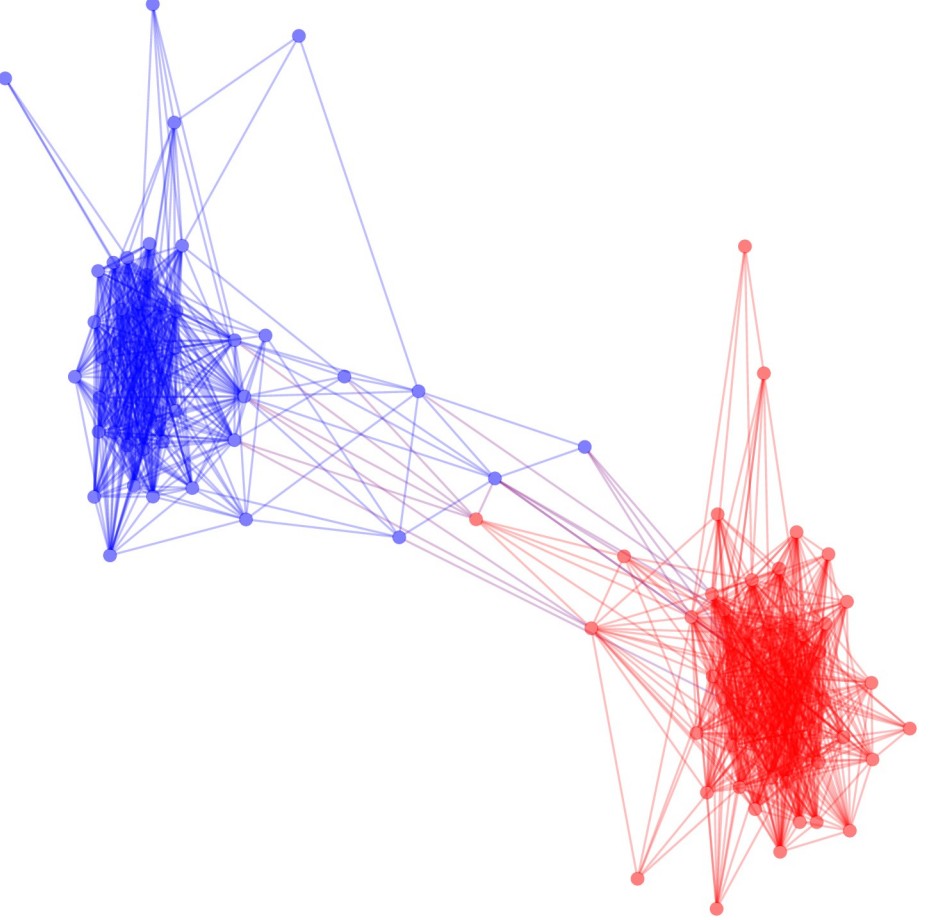

**Fig 4. The positive backbone of the US Senate co-sponsorship network under the stochastic degree sequence model.**

The function can also save each value of $G_{ij}^*$ for a given $i, j$. This is useful for visualizing an example of the empirical null edge weight distribution generated by the model. The values $i, j$ correspond to the row and column indices of a cell in the projected matrix and can be input as either numeric values or a string containing the row names. These values are returned in the list `dyad_values`.

Using the fixed degree sequence model on the senate data set will allow us to compare our observed values to a distribution where each senator sponsors the exact same number of bills and each bill is sponsored by the exact same number of people. We can find the backbone using the fixed degree sequence model as follows:

```
> fdsm <- fdsm(senate, trials = 1000, dyad = c("Booker, C.
(NJ-D)",
         "Warren, E. (MA-D)"))
Approximating the distribution using Curveball FDSM
Estimated time to complete is 81 secs
```

The `dyad_values` output is a list of the $G_{ij}^*$ values for each of the 1000 trials, where $i$ = "Booker, C. (NJ-D)" and $j$ = "Warren, E. (MA-D)". These values correspond to the number of bills Senators Booker and Warren would be expected to co-sponsor when we create a random bipartite graph with the curveball algorithm where: (a) the number of bills sponsored by Senator Booker, by Senator Warren, and all other Senators was fixed, and (b) the number of senators sponsoring each bill was fixed. We can compare their actual number of co-sponsorships, 98, to what is generated under our null model. We can view a histogram of the expected co-sponsorships generated in each of the 1000 trials as follows (see Fig 5):

```
> hist(fdsm$dyadvalues, freq = FALSE, xlab = "Number of Co-
Sponsorships")
> lines(density(fdsm$dyadvalues))
```

To extract the backbone, we supply the `backbone.extract()` function with the proportion matrices `positive` and `negative`.

```
> fdsmbb <- backbone.extract(fdsm, alpha = 0.01, signed = TRUE)
```

The FDSM backbone, based on 1000 Monte Carlo samples, requires approximately 81 seconds to extract. Using the fixed degree sequence model allows us to see more of the partisan structure we assume to be present in the United States Senate in Fig 6. This expected partisan structure is confirmed by the backbone's high modularity ($M$ = 0.468).

## Backbone model comparison and selection

As the above examples illustrate, the `backbone` package can be used to extract several different backbones using different backbone models. Table 1 compares the five backbones extracted above, in terms of running time, modularity, and structural similarity (expressed as a Pearson correlation coefficient).

The universal and hypergeometric backbones are very fast to extract because they rely only on arithmetic and straightforward parametric distributions. In contrast, the FDSM is relatively slow because it must empirically approximate edge weight probability distributions using Monte Carlo methods and repeated matrix multiplication. The SDSM occupies a middle-ground in computational complexity because, while it does not require costly Monte Carlo methods or repeated matrix multiplication, it does involve estimation of cellwise probabilities and a more complex parametric distribution (i.e. Poisson-Binomial).

Accuracy of these backbones is difficult to assess directly, however because the US Senate is known to be polarized along political party lines, the magnitude of the backbone's modularity when partitioned by political party provides an indicator of the extent to which each backbone

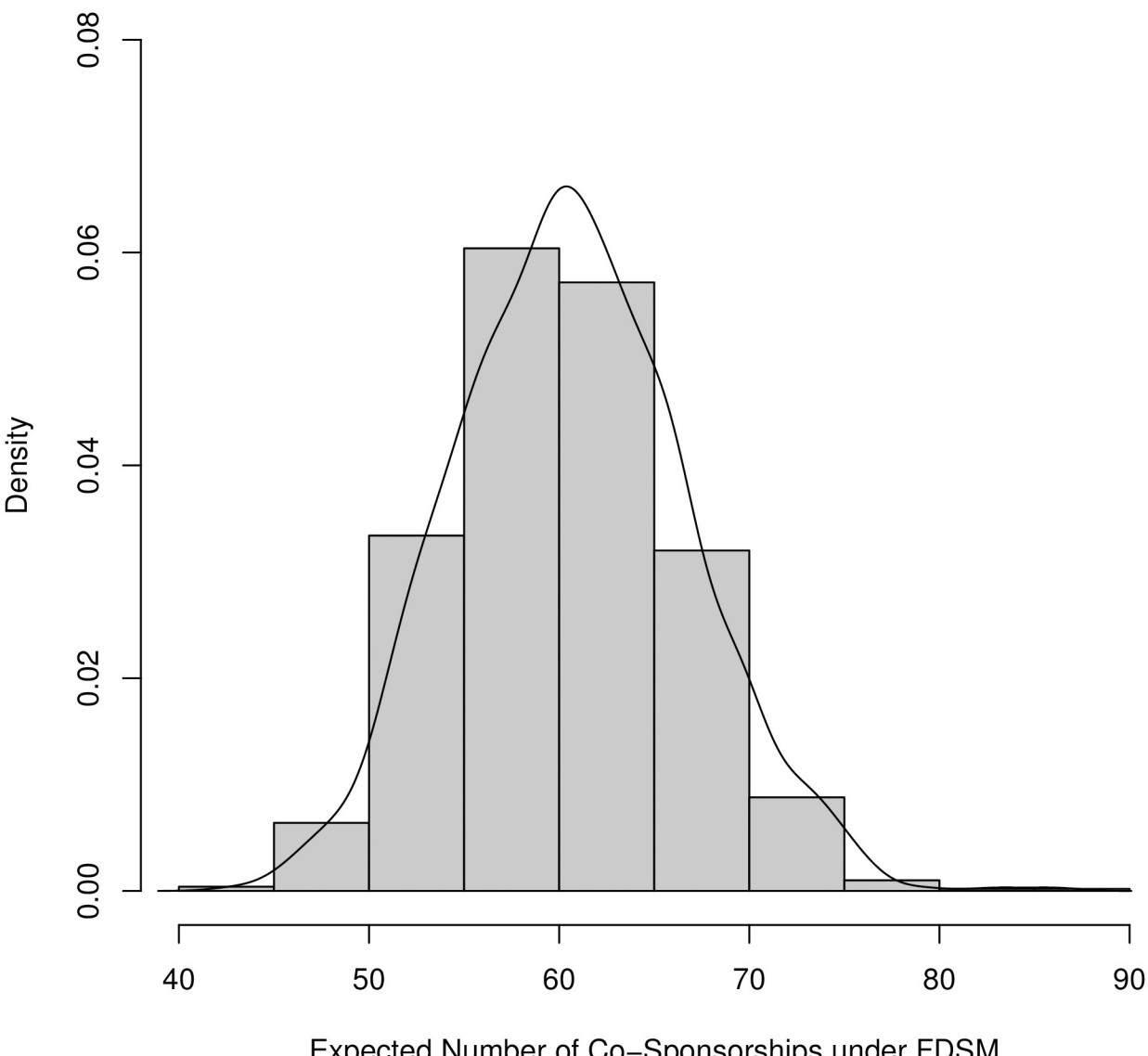

**Fig 5. A histogram of the expected co-sponsorships between Senators Cory Booker and Elizabeth Warren under the fixed degree sequence model (1000 samples).** A positive edge between Booker and Warren would be preserved in the FDSM backbone because their actual number of co-sponsorships (98) is statistically significantly larger.

reproduces this known structure. A universal threshold backbone where $T = 0$ fails to capture this polarized structure at all, while a universal threshold backbone where $T = M + SD$ and a hypergeometric backbone begin to capture this structure. However, the SDSM and FDSM backbones are best able to reproduce the known partisan polarization in the US Senate.

Examining the Pearson correlation coefficient computed on the vectorized adjacency matrices of these backbones quantifies their pairwise structural similarity. The universal threshold backbone where $T = 0$ is minimally correlated with any other backbone, reflecting the fact that its very dense structure is unlike the other sparser backbones. The universal threshold backbone where $T = M + SD$ shares some edges in common with backbones generated using HM, SDSM, and FDSM ($r = 0.47 − 0.51$), and the hypergeometric backbone is even

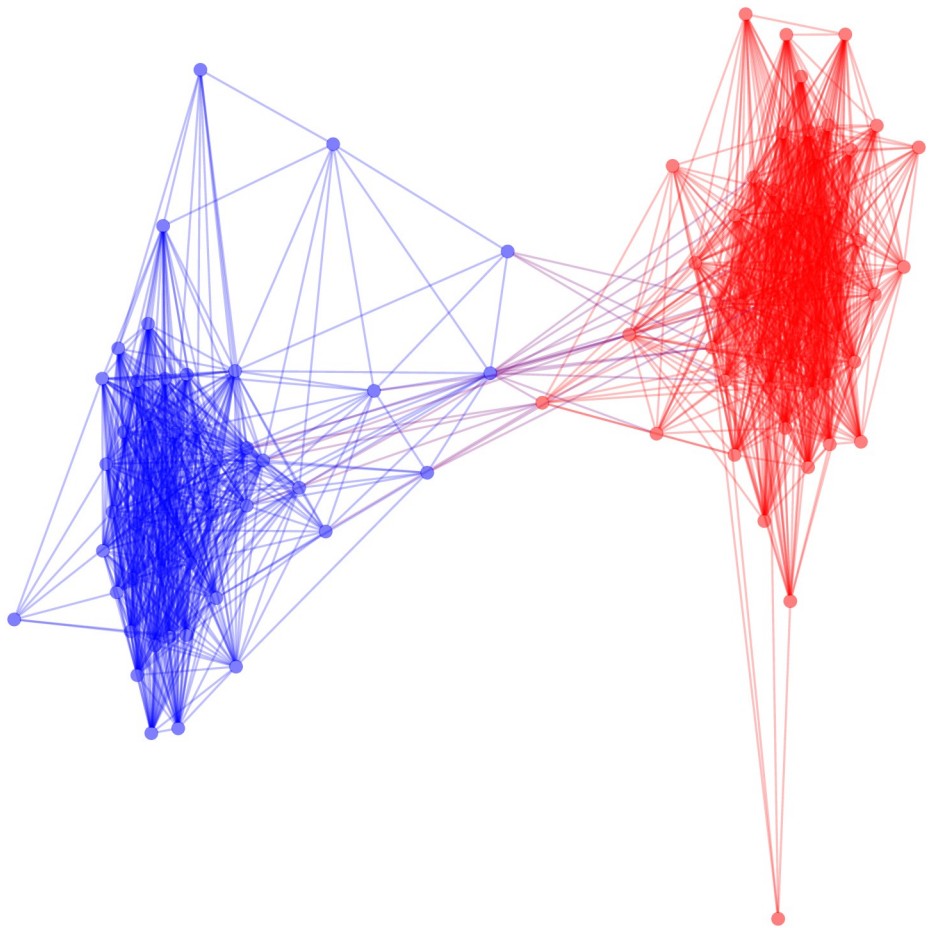

**Fig 6. The positive backbone of the US Senate co-sponsorship network under the fixed degree sequence model.**

more similar to those generated by SDSM and FDSM ($r = 0.71 - 0.73$). Notably, the SDSM and FDSM backbones are nearly identical ($r = 0.96$).

Collectively, these comparisons suggest that SDSM performs best in the US Senate example: it accurately reproduces the known polarized structure, but more quickly than FDSM. However, specific characteristics of the bipartite data should guide backbone extraction model selection in other contexts. First, because it imposes no controls on any characteristics of the data, the *universal threshold* model is suitable only when the row sums of the bipartite matrix

**Table 1. Comparison of US Senate backbones.**

| Backbone model | Time[a] | Modularity[b] | Pearson correlation | | | | |
|---|---|---|---|---|---|---|---|
| Universal ($T = 0$) | 0.01 | -0.005 | — | | | | |
| Universal ($T = M + SD$) | 0.01 | 0.277 | 0.01 | — | | | |
| Hypergeometric | 0.02 | 0.215 | 0.14 | 0.47 | — | | |
| SDSM | 45 | 0.471 | -0.01 | 0.51 | 0.71 | — | |
| FDSM (1000 samples) | 81 | 0.468 | -0.00 | 0.50 | 0.73 | 0.96 | — |

[a] In seconds, using a 2.3 Ghz Quad-Core Intel Core i7.
[b] Based on a partition by political party affiliation.

exhibit limited variation or this variation is not theoretically meaningful *and* the column sums of the bipartite matrix exhibit limited variation or this variation is not theoretically meaningful. These conditions are unlikely to occur in practice, however in cases where the universal threshold model is appropriate, the threshold value should be chosen based on theory. Second, because it imposes controls on the row sums only, the *hypergeometric* model is suitable when the row sums of the bipartite matrix exhibit meaningful variation, but the column sums do not. Finally, because they impose controls on both the row and column sums, the *stochastic degree sequence model* and *fixed degree sequence model* are suitable when both the row sums and the column sums of the bipartite matrix exhibit meaningful variation. This is likely to be the most common case for most empirical bipartite data.

As the comparisons in Table 1 illustrate, the SDSM may be regarded as a fast approximation of the FDSM. Its speed derives from the fact that it only approximately controls for row and column sums, while FDSM exactly controls for row and column sums. Therefore, the choice between SDSM and FDSM depends on two related practical considerations. First, it depends on the size of the data: except for relatively small bipartite data, FDSM will often be impractically slow. Second, it depends on the desired precision of the *p*-values quantifying each edge's statistical significance, which for FDSM are constrained by the number of Monte Carlo samples to 1/samples. This can be problematic when the familywise error rate is controlled using a Bonferroni or Holm-Bonferroni correction because these corrections require high-precision *p*-values. For example, in the Senate network there are 4950 edges that must be tested during backbone extraction, and thus 4950 independent statistical tests. To achieve statistical significance at a familywise error rate of 0.05 via a Bonferroni correlation, an edge's statistical significance must be $p < 0.00001$. Achieving this level of precision among *p*-values using FDSM would require drawing at least 100,000 Monte Carlo samples. Therefore, if a specific familywise error rate is desired, FDSM will be impractically slow even for relatively small bipartite data. For these reasons, while further research is needed to determine the quality of SDSM's approximation of FDSM, SDSM will often be a suitable backbone extraction model choice.

## Discussion

We have presented four methods—universal threshold, HM, SDSM, and FDSM—for identifying significant links in a weighted bipartite projection, and thus for extracting its binary or signed backbone. We have also introduced and demonstrated the `R backbone` package, which implements these methods in a common framework facilitating their use. Together, the methods and package offer ways for researchers to reduce the structural artifacts (e.g., excessive density and clustering) and complexity of bipartite projections, making them easier to analyze and visualize. This is likely to be most useful in cases where the researcher is interested in an unobserved unipartite network that is not simply a bipartite projection of something (e.g., a social network), but that can be inferred (even with some error) from observed bipartite data (e.g., co-behaviors).

There are two important cases where `backbone` is not appropriate and should *not* be used. First, if the research question can be answered by directly analyzing the original bipartite graph, or the central message can be communicated by visualizing the original bipartite graph, then it is not necessary and indeed may be misleading to examine a bipartite projection or its backbone. In these cases, researchers should instead use tools designed for the analysis of bipartite networks, including for example bipartite extensions of ERGM [37] and SIENA [38]. Second, it is possible to obtain a weighted (via projection) or unweighted (via `backbone`) square symmetric matrix from *any* bipartite data, however this does not imply that the matrix can be meaningfully analyzed as a network. Treating a square symmetric matrix obtained via

projection or backbone extraction as a network requires an *explicit conceptual* rationale for why the sharing of artifacts by a pair of agents provides information about those agents' relationship with each other, and a description of what kind of relationship this is. In some cases, the rationale may be simple and straightforward: social events and other venues offer opportunities for people to meet and interact, so observing two people attending many of the same events may indirectly provide information about the possibility of a social relationship between them [1]. In other cases—for example, psychological networks in which clinical symptoms (the 'agents') are linked because they co-occur in patients (the 'artifacts') [39]—the rationale is less obvious [40, 41].

Even when backbones of bipartite projections are used in appropriate ways, there remain a number of open questions that should guide future research in this area and future development of the `backbone` package. First, although the `backbone` package implements four methods for extracting the backbone of bipartite projections, we know little about how the backbones generated by these methods differ from one another. We have briefly explored a comparison of the backbones generated by these methods in the context of the US Senate, but more formal comparisons and empirical comparisons in other contexts are necessary. Second, to the extent that these methods are intended to provide insight into an unobserved but underlying unipartite network, we know little about which backbone extraction methods do so most accurately. Future research should develop methods for establishing the validity of these backbone methods, for example, by comparing extracted backbones to ground-truth unipartite networks. While these remain open questions, the availability of the `backbone` package makes it possible to begin answering them, and also provides a platform for implementing methods and exploring similar questions in the more general case of extracting the backbone of non-projection weighted graphs.

## Supporting information

**S1 File.**
(ZIP)

## Author Contributions

**Conceptualization:** Rachel Domagalski, Zachary P. Neal, Bruce Sagan.

**Formal analysis:** Rachel Domagalski, Zachary P. Neal.

**Methodology:** Rachel Domagalski, Zachary P. Neal.

**Project administration:** Zachary P. Neal.

**Resources:** Zachary P. Neal, Bruce Sagan.

**Software:** Rachel Domagalski, Zachary P. Neal.

**Supervision:** Zachary P. Neal, Bruce Sagan.

**Visualization:** Zachary P. Neal.

**Writing – original draft:** Rachel Domagalski, Zachary P. Neal.

**Writing – review & editing:** Rachel Domagalski, Zachary P. Neal, Bruce Sagan.

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
