## [Decision Letter · Decision Letter 0]

5 Nov 2020

PONE-D-20-23636

backbone: An R package for extracting the backbone of bipartite projections

PLOS ONE

Dear Dr. Neal,

Thank you for submitting your manuscript to PLOS ONE. After careful consideration, we feel that it has merit but does not fully meet PLOS ONE’s publication criteria as it currently stands. Therefore, we invite you to submit a revised version of the manuscript that addresses the points raised during the review process.

The 3 reviewers are very enthusiastic with the paper that fill the gap of explaining a very useful new tool. The paper fitts very well to the journal because this tool could be used by many disciplines and deserves a high visibility. Some interesting and constructing remarks from reviewers 2 and 3 could help to enhance the clarity of the paper.... These emarks are not fundamental, and you can consider that once these remarks will be addressed, I will directly accept the publication without necessitating a second round of reviewing (so minor revision - close to accepted).

We look forward to receiving your revised manuscript.

Kind regards,

Celine Rozenblat

Academic Editor

PLOS ONE

Journal Requirements:

Reviewers' comments:

Reviewer's Responses to Questions

**Comments to the Author**

1. Is the manuscript technically sound, and do the data support the conclusions?

Reviewer #1: Yes

Reviewer #2: Yes

Reviewer #3: Yes

2. Has the statistical analysis been performed appropriately and rigorously? 

Reviewer #1: Yes

Reviewer #2: Yes

Reviewer #3: Yes

3. Have the authors made all data underlying the findings in their manuscript fully available?

Reviewer #1: Yes

Reviewer #2: Yes

Reviewer #3: Yes

4. Is the manuscript presented in an intelligible fashion and written in standard English?

Reviewer #1: Yes

Reviewer #2: Yes

Reviewer #3: Yes

5. Review Comments to the Author

Reviewer #1: The paper introduces the backbone package for R, which includes several methods to infer a binary one-mode projection from a two-mode network.

The paper is well organized and explains the package structure well. The most important part of the paper, though, is given in the last paragraph (line 424-436). None of the implemented backbone extraction methods can be considered superior and there are, so far, no theoretical or technical guidelines for which method to choose in which context. The package collates the most important backbone extraction methods from the literature and makes them available in a way that is useful to (hopefully) answer this question in the near future.

Typo:

Line 143: paramenters->parameters

Reviewer #2: This paper is a continuation of a previous paper that presented several methods for extracting backbones from bipartite graphs (Neal, 2014). It introduces an R package to implement these methods and gives a specific example of the application of these methods in the case of bill co-sponsorship relations among Senators in the 114th session of the United States Senate.

The article also suggests refinements from a mathematical point of view:

In particular, it proposes to design polytope-based probabilities to implement the stochastic degree sequence model.

It also defends the choice of the curveball algorithm to generate fixed row and column sum random matrices (FDSM model).

We salute the quality of the writing and the clarity of the demonstrations.

We propose only slight modifications, make a few suggestions and encourage the authors to go a little further in analysing the differences between the different methods proposed and the different backbones obtained. To do so, we encourage them to use the question posed in the example: "how many bills do two senators have to co-sponsor before we can conclude that they are political collaborators"?

Here are the details of our review that can also be found in the annotated version of the manuscript (some additional minor comments are present in the annotated version):

Line 16: “when transforming a bipartite graph into a unipartite graph via projection, information about

the `events' responsible for edges between vertices is lost”  This is not entirely true since one can partly keep this information by weighting the links accordingly: see https://toreopsahl.com/tnet/two-mode-networks/projection/

Line 21-24: “observing two people attending many of the same events (i.e. a large edge weight) may still be uninteresting if they each attended many large events, but observing two people attend just a few of the same events (i.e. a small edge weight) may be interesting if they each attended only a few small events” 

Weighting methods such as the ones mentioned in the link above offer to deal with such an issue. To what extend would you say that backbones extraction offer a better way to cope with this?

Line 61-65: “Bipartite projections are of interest in social network analysis because they allow us to construct a network from artifact affiliations, which are often easier to obtain than taking a survey of the network members. If the number of members of the network is large, getting complete and reliable information regarding relationships between members can involve long and repetitive survey techniques which can lead to survey fatigue and costly field work.” 

Note that it is true for social network analysis in sociology... but in history, for instance this possibility (taking a survey of the network members) is not even possible.

Line 70-71: “Because bipartite projections are weighted, and because what counts as a `large' or `small' weight can differ for each edge, it can be useful to reduce this information” 

I understand that you refer to the idea developed in the introduction regarding the difference between large and small events in the case of events co-attendance, but maybe the wording could be more explicit here, perhaps with an example.

Line 136-137: “That is, for any given B*, each row and column sum may be higher or lower than its observed degree in B, but on average they are equal” 

Again, I'm not sure that the formulation is clear enough here, it seems to me that this sentence can be understood in different ways.

Line 143: typo  “paramenters” instead of “parameters”

Line 193-195: “In visualizations of the extracted backbones, we depict Republican senators by red vertices, Democratic senators by blue vertices, and Independent senators by green vertices.”  Actually, we can’t see any green on the figures.

In the Replication Code available on the website https://osf.io/r9gvh/, we see that democrats and independents are plotted in the same color because of this line (l. 16):

V(iB)$party[V(iB)$party == "I"] <- "D"

Line 196: “Although we discuss signed backbones in the text, for visual clarity only provide figures for binary backbones which contain positive edges.”  missing “we” before “provide”

Line 196-198: “Positive relations of collaboration between two Republicans are red, between two Democrats are blue, and for all other pairs are gray”  the other pairs look rather purple to me

Line 235-239: “The differences in the number of bills co-sponsored prompts an important underlying question: how many bills do two senators have to co-sponsor before we would be justified in concluding they are political collaborators? Similarly, how few bills do they have to co-sponsor before we would be justified in concluding they are political opponents? These questions are what the backbone package seeks to answer.” 

In addition to plotting the backbones resulting from the different methods, it would be interesting to find a way to compare the answers that each extraction method can possibly give to these questions in the case of the 114th session of the US Senate. This could contribute to respond to the more general question addressed in the discussion part of the paper: how can we know which extraction method is the more relevant for the research question we ask?

Line 356-357: “We are able to see more of the partisan structure that is suggested to be present in the US Senate in fig. 4”

Line 392-393: “Using the fixed degree sequence model allows us to see more of the partisan structure we assume to be present in the United States Senate in fig. 6.” 

The argument given to comment on Figures 5 and 6 is identical, which is rather disappointing for interpreting the difference between the results obtained using the two distinct methods.

To go further, could you give more details on the difference between this backbone (diplayed on fig. 6) and the previous one (displayed on fig. 4), and try answering the following question: which one performs the most in displaying the expected partisan structure? And why is it so?

Line 430-431: “Second, to the extent that these methods are intended to provide insight into an unobserved but underlying unipartite network, we know little about which backbone extraction methods do so most accurately.” 

You could start giving us some clues about this using the example of bill co-sponsorships.

Reference 18: Error in the volume number, it is volume 30 and not volume 9.

Reviewer #3: The paper stands very well as it is -- it is clear, straightforward, and rich in details on the software implementation of the method, which will help others to build on it.

I attended the authors' presentation at Sunbelt 2020, during which they presented almost exactly the same content, also with great clarity and success.

I will add to that that I have successfully used the package documented in the paper myself, and that I have successfully rebuilt the package locally, including all unit tests and vignettes, on top of running the replication code for the article.

As such, the package validates the reviewing principles used by software-oriented journals like JOSS or the Journal of Stat. Software, where the paper would also fit rather well.

A few more remarks:

1.

The paper states to implement the "most common" (page 2) bipartite backbone extraction methods, but the commonality of the methods covered by the package is undocumented in the paper.

Perhaps the paper should more readily redirect the reader to ref. 9 (Neal 2014), in order to clearly indicate that the commonality of the methods covered by the package is demonstrated in this paper.

2.

The paper does not document parameter tuning, i.e. how much of the total weights, nodes and edges are preserved in the backbone when parameter alpha is lowered from e.g. 0.05 to 0.001.

A good example of that is found in Serrano et al. 2009 (ref. 30 in the present paper). The same paper also documents how that parameter influences the network topology of the resulting backbone.

Doing so would enhance the example part of the paper, which is weak in comparison to the other (very good) papers published by the main author on the topic of Congressional political polarization measured through bill co-sponsorship (in Sci. Rep. 2020 and in Soc. Net. 2020).

3.

In line with the previous remark, it would be helpful to further compare how each method differs from each other from a computational perspective.

The only part of the paper where this appears is when the authors mention the computational cost of the FSDM (page 6). This point should itself be further documented, e.g. by giving some idea of how long it would take to apply that method on a much larger network, since much larger networks are very common in complex network analysis.

It might also be useful to show that the other methods documented in the paper remain computationally unexpensive on mugh higher network sizes and under varying densities, uness one considers this to be trivial enough to be left out.

* * *

This paper fits well with the others published or co-published by the main author, which I mentioned above.

Congratulations to the authors for their work, which is a very helpful addition to existing research on the same topic.

6. PLOS authors have the option to publish the peer review history of their article (what does this mean?). If published, this will include your full peer review and any attached files.

Reviewer #1: No

Reviewer #2: No

Reviewer #3: **Yes: **François Briatte

---

## [Decision Letter · Decision Letter 1]

9 Dec 2020

backbone: An R package for extracting the backbone of bipartite projections

PONE-D-20-23636R1

Dear Dr. Neal,

We’re pleased to inform you that your manuscript has been judged scientifically suitable for publication and will be formally accepted for publication once it meets all outstanding technical requirements.

Kind regards,

Celine Rozenblat

Academic Editor

PLOS ONE

Additional Editor Comments (optional):

Thanks for the authors to have addressed all the requested minor changes. As said before, at this condition, it is now ready for publication.

Reviewers' comments:

Reviewer's Responses to Questions

**Comments to the Author**

1. If the authors have adequately addressed your comments raised in a previous round of review and you feel that this manuscript is now acceptable for publication, you may indicate that here to bypass the “Comments to the Author” section, enter your conflict of interest statement in the “Confidential to Editor” section, and submit your "Accept" recommendation.

Reviewer #3: All comments have been addressed

2. Is the manuscript technically sound, and do the data support the conclusions?

Reviewer #3: Yes

3. Has the statistical analysis been performed appropriately and rigorously? 

Reviewer #3: Yes

4. Have the authors made all data underlying the findings in their manuscript fully available?

Reviewer #3: Yes

5. Is the manuscript presented in an intelligible fashion and written in standard English?

Reviewer #3: Yes

6. Review Comments to the Author

Reviewer #3: Thanks to the authors for replying to Comments 4.2 and 4.3 as they have. In particular, I understand that the authors have reserved parameter tuning for another paper (Comment 4.3 and reply), which certainly makes sense, and which they might want to announce in the present paper. Looking forward to reading the follow-up!

7. PLOS authors have the option to publish the peer review history of their article (what does this mean?). If published, this will include your full peer review and any attached files.

Reviewer #3: **Yes: **François Briatte

---

## [Editor Report · Acceptance letter]

18 Dec 2020

PONE-D-20-23636R1 

backbone: An R package for extracting the backbone ofbipartite projections 

Dear Dr. Neal:

I'm pleased to inform you that your manuscript has been deemed suitable for publication in PLOS ONE. Congratulations! Your manuscript is now with our production department. 

Kind regards, 

on behalf of

Prof. Celine Rozenblat 

Academic Editor

PLOS ONE